# Tree Physiological Variables as a Proxy of Heavy Metal and Platinum Group Elements Pollution in Urban Areas

**DOI:** 10.3390/biology12091180

**Published:** 2023-08-29

**Authors:** Zulema Varela, Javier Martínez-Abaigar, Rafael Tomás-Las-Heras, José Ángel Fernández, María-Ángeles Del-Castillo-Alonso, Encarnación Núñez-Olivera

**Affiliations:** 1CRETUS, Ecology Unit, Department Functional Biology, Faculty of Biology, Universidade de Santiago de Compostela, 15782 Santiago de Compostela, Spain; jangel.fernandez@usc.es; 2Faculty of Science and Technology, University of La Rioja, 26006 Logroño, Spain; javier.martinez@unirioja.es (J.M.-A.); rafael.tomas@unirioja.es (R.T.-L.-H.); maria-angeles-del.castillo@unirioja.es (M.-Á.D.-C.-A.); encarnacion.nunez@unirioja.es (E.N.-O.)

**Keywords:** biomonitoring, catalytic converters, *Ligustrum lucidum*, palladium, rhodium

## Abstract

**Simple Summary:**

Air pollution in urban areas represents a major environmental risk to human and ecosystem health. Our aim was to test the adequacy of the *Ligustrum lucidum* physiological variables as proxies for heavy metal and platinum group element pollution. Chlorophyll, nitrogen and F_v_/F_m_ generally showed high values typical of healthy plants, and they did not seem to be consistently affected by the different pollutants. Regarding flavonoid content, it was negatively correlated with heavy metals, which did not confirm its role as a protectant against metal stress. The relatively low levels of pollution usually found in the city of Logroño, together with the influence of other environmental factors and the relative tolerance of *Ligustrum lucidum* to modest atmospheric pollution, probably determined the only slight response of physiological variables to heavy metals and platinum group elements.

**Abstract:**

Physiological variables (the content of chlorophyll, flavonoids and nitrogen, together with F_v_/F_m_) and the content of ten heavy metals (As, Cd, Cu, Hg, Mn, Ni, Pb, Sb, V and Zn) and two platinum group elements (PGEs: Pd and Rh) were measured in the leaves of 50 individuals of *Ligustrum lucidum* trees regularly distributed in the city of Logroño (Northern Spain). Three of these variables increased with increasing physiological vitality (chlorophyll, nitrogen and F_v_/F_m_), whereas flavonoids increased in response to different abiotic stresses, including pollution. Our aim was to test their adequacy as proxies for the pollution due to heavy metals and PGEs. The three vitality indicators generally showed high values typical of healthy plants, and they did not seem to be consistently affected by the different pollutants. In fact, the three vitality variables were positively correlated with the first factor of a PCA that was dominated by heavy metals (mainly Pb, but also Sb, V and Ni). In addition, F_v_/F_m_ was negatively correlated with the second factor of the PCA, which was dominated by PGEs, but the trees showing F_v_/F_m_ values below the damage threshold did not coincide with those showing high PGE content. Regarding flavonoid content, it was negatively correlated with PCA factors dominated by heavy metals, which did not confirm its role as a protectant against metal stress. The relatively low levels of pollution usually found in the city of Logroño, together with the influence of other environmental factors and the relative tolerance of *Ligustrum lucidum* to modest atmospheric pollution, probably determined the only slight response of the physiological variables to heavy metals and PGEs.

## 1. Introduction

Air pollution is perceived as the second greatest environmental concern among European citizens, after climate change [1]. This pollution has a significant impact on the health and economy of Europe’s urban populations, shortening citizens’ lives, increasing health costs and reducing productivity due to lost working time, often associated with heart problems, respiratory diseases and lung cancers [2,3]. Although air pollution affects the entire population, people living in urban areas or close to industrial areas or high-traffic roads are more exposed to this harm because urban and peri-urban areas are a major source of the emissions of, for example, heavy metals into air [3]. They are toxic to biota, and although their ambient air concentrations only exceed legal thresholds in a few areas in Europe, the atmospheric deposition of heavy metals leads to the exposure of ecosystems and organisms to these harmful pollutants, which can also bioaccumulate in the food chain, with detrimental effects on human health [4,5]. Urban areas are also the main sources of emissions of naturally rare non-essential metals such as palladium (Pd) and rhodium (Rh), which belong to platinum group elements (PGEs). These elements have been widely used in the manufacture of automotive catalysts that catalyze and control harmful exhaust emissions since the 1980s [6,7]. They are predominantly emitted as nanoparticles in metallic or oxide form, which are readily dissolved and can easily be converted into bioavailable complexes, being therefore another major environmental hazard.

The use of ornamental trees for the biomonitoring of urban air quality is a widely used technique to provide up-to-date information on the levels of pollutants in cities [8,9]. The privet *Ligustrum lucidum*, an evergreen tree very common in urban landscaping, is frequently used as a passive biomonitor for PM_10_/PM_2.5_, PAHs, persistent and emerging semi-volatile organic pollutants and heavy metals [10,11,12,13,14,15,16]. Its leaves are covered by a cuticular layer of wax, conferring excellent properties to capture atmospheric pollutants, mainly in the form of particles [17]. In addition, some authors have reported that *L. lucidum* is more sensitive than other species to atmospheric pollution, and this sensitivity is manifested in its physiology as a decrease in total chlorophyll content, ascorbic acid, stomatal density or water content [11,15,18]. Therefore, it is recommendable to use its physiological variables to monitor atmospheric pollution and the potential concomitant damage caused to the tree and, consequently, to the health of cities and human beings.

In general, physiological measurements of tree leaves can be used to assess the damage caused by the exposure to adverse environmental conditions and, presumably, pollutants [19,20,21]. Some variables commonly used in these studies are the maximum quantum yield of photosystem II (F_v_/F_m_), chlorophyll content and flavonoid content. Regarding F_v_/F_m_, it estimates the leaf photosynthetic performance, which in turn is indicative of the leaf vitality [22]. Thus, the higher the F_v_/F_m_, the better the physiological state of the plant. Nevertheless, F_v_/F_m_ can respond to many different environmental factors, such as excess light, cold, heat, drought, excess water, nutritional deficiencies, pollution, etc., and this may mask the response to a specific individual factor under natural conditions. Chlorophyll content is also somewhat related to the leaf’s ability to photosynthesize, and therefore to its physiological vitality, and can be interpreted in the same way as F_v_/F_m_: the higher the chlorophyll content, the higher the physiological vitality. Hence, a decrease in chlorophyll content may indicate the negative influence of different adverse factors [23,24]. Similarly, given that nitrogen (N) is an essential element for the plant, and it is subjected to a certain metabolic control [25,26], its content could also be used as a physiological indicator of vitality. However, the information available on the relation between N content and metal pollution is scarce, including whether metal pollution is connected in some way with the sources of N emissions and a stable N isotopic ratio δ^15^N. Finally, flavonoids are a class of secondary metabolites, many of which possess antioxidant functions, which the plant uses in response to or to protect itself from abiotic and biotic adverse factors, such as excess light, ultraviolet radiation, oxidative damage, nutritional deficiencies, injuries, herbivores, etc. [27,28,29,30]. Therefore, higher flavonoid content generally indicates that the plant is responding to some stressor in order to avoid, as far as possible, the associated physiological damage.

Despite the relatively frequent use of *L. lucidum* in biomonitoring studies, only a few have directly correlated the concentrations of pollutants in its leaves with its physiological characteristics, trying to elucidate whether the potential physiological damage is due to contamination or to any other environmental stress factor. With this in mind, the aim of this study was to test whether several physiological variables measured in *L. lucidum* leaves (the content of chlorophyll, flavonoids and N, together with F_v_/F_m_) could directly be related to the leaf content of heavy metals and PGEs (Pd and Rh). If so, physiological variables could be used as proxies for the pollution caused by these elements in urban environments.

## 2. Materials and Methods

### 2.1. Sampling and Processing

A census of ornamental privets of the species *Ligustrum lucidum* W.T. Aiton was carried out throughout the city of Logroño (Spain, 42°28′12″ N latitude, 2°26′44″ W longitude, 384 m altitude; ca. 150,000 inhabitants). This census amounted to a total of 2957 individuals, from which 50 trees distributed in an approximate 500 m grid within the city were selected for analysis (Figure 1). Only healthy and robust specimens of privet were used, all of them of the varieties with green leaves, although, in some cases, to cover gaps, some specimens with variegated leaves had to be used. 

Leaves were collected from 29 to 31 May 2018. In each selected tree, four branches were cut in the N, S, E and W orientations. The branches were placed in airtight plastic bags and transported to the laboratory. Once in the laboratory, and on the same day as collection, the recently sprouted spring leaves were discarded, and two well-developed basal leaves of each branch were selected. Non-destructive physiological measurements were performed on these leaves (see Section 2.2). Then, the leaves were oven-dried (24 h at 40 °C) and pulverized in a tangential mill (Restch MM-400, with zirconium oxide vessels) to obtain a composite sample for analysis.

### 2.2. Physiological Measurements

Measurements were carried out on eight leaves from the four orientations of the trees (two leaves for each orientation), using the central part of the leaf blade and avoiding the main nerves. All the variables were measured following [31]. Chlorophyll fluorescence (the maximum quantum yield of photosystem II, F_v_/F_m_) was determined using a modulated fluorimeter (PAM-2500, Walz, Effeltrich, Germany) following the saturating pulse methodology. The chlorophyll and flavonoid content of leaves were also determined, using the non-invasive Dualex^®^ 4 SCIENTIFIC leaf clamp sensor (FORCE-A, Paris, France). First, Dualex measurements were performed, and then leaves were adapted to the dark to measure F_v_/F_m_.

### 2.3. Chemical Analysis

Aliquots of 0.2 g of privet leaves from each of the composite samples were digested with 8 mL HNO_3_ and 2 mL H_2_O_2_ (30%) in a microwave oven (Ethos-1, Milestone, Brondby, Denmark) in Teflon vessels at high pressure, increasing the temperature to 190°C for 25 min and maintaining this temperature for 15 min. The cold extracts were then brought to a final volume of 25 mL with MilliQ water. The concentrations of As, Cd, Ni, Pb, Cu, Sb, Pd, Rh and V in the extracts were determined spectrometrically on an ICP-MS (Agilent 7700x, Santa Clara, CA, USA), while those of Mn and Zn were determined by AAS (Pekin Elmer 2100, Waltham, MA, USA). In the case of Cd, the samples that had concentrations below the limit of quantification were redetermined by GFAAS (Perkin Elmer AAnalyst 600), after suspension with ultrasound extraction. In the case of Hg, an elemental analyzer (Milestone DMA 80) was used in which the homogenized and dried material was directly introduced. Analytical quality control was performed by parallel analysis of (i) blanks to calculate the limit of quantification of the technique used for each of the elements (1 per 9 samples); (ii) certified reference materials using matrices similar to those analyzed (1 per 9 samples); and (iii) analytical duplicates to verify the replicability of the determinations performed (1 per 9 samples). The overall errors obtained for metals were less than 10%, conferring an adequate level of analytical quality.

For the determination of total N, aliquots of 3 ± 0.1 mg of dry material were weighed on a precision balance (Mettler Toledo XP26, Singapore) and packed in tin capsules (Eurovector, Pavia, Italy). The capsules were then analyzed on an N elemental analyzer (FlashEA1112 ThermoFinnigan, San Jose, CA, USA). To determine the stable N isotopic ratio (δ^15^N), the elemental analyzer was coupled to an isotope ratio mass spectrometer (Deltaplus, ThermoFinnigan). Calibration of the reference gas (N_2_) for atmospheric ^15^N was performed with IAEA-N-1 ((NH_4_)_2_SO_4_), IAEA-N-2 ((NH_4_)_2_SO_4_) and IAEA-NO-3 (KNO_3_) as standards. The isotope ratios (^15^N/^14^N) of the samples were checked against the standard (atmospheric N_2_), so that the ratios obtained were comparable. The relative abundance of ^15^N in the sample (δ^15^N) was calculated using the expression δ^15^N (‰) = [(Rsample/Rstandard) − 1] × 10^3^, where R is the ^15^N/^14^N ratio. The global error determined using analytical replicates was 8%.

### 2.4. Data Analysis

A new probabilistic method proposed by Giráldez et al. (2022) [32] and based on a Gaussian mixture model was used to determine whether the levels of elements present in the leaves could be considered pollution or not. The method categorizes all observations studied as a “background level” and, subsequently, spatial statistical techniques are applied to determine the probability of exceeding this baseline level. In this way, the background level considered would correspond to unpolluted sites, whereas higher levels would represent polluted sites. To relate this contamination to physiology, first, a factor analysis was performed for metals and PGEs and, with the factor scores of each observation, a Spearman’s test and a principal component analysis (PCA) were performed to see if there was a correlation between elements and physiological variables. All the analyses were performed with the R-3.4.0 statistical software [33].

## 3. Results

The medians of the 12 elements analyzed (metals and PGEs) are shown in Table 1. The essential and major metals that are widely distributed naturally had the highest medians (Mn > Zn > Cu), followed by elements that have no known biological functions and only act as contaminants: Ni > Pb > V > Pd > As > Sb > Cd > Hg > Rh. In general, there did not seem to be a high level of pollution, on the basis of the 50 trees studied (Figure 2), because only 19 of them showed contamination for one element, four for two elements (trees 11, 16, 23 and 46), two for three elements (trees 14 and 32) and only one tree for four elements (number 6). These more enriched trees were found distributed throughout the city, mainly in the northern and western areas but with some presence in other sectors. There were more contaminated trees for Zn, Pb, and Ni than for the remaining elements, and no tree displayed Mn and V pollution. Regarding the PGEs, they seemed to behave in a similar way since the trees polluted by these elements were practically the same (e.g., 11, 32 and 46).

Regarding non-destructive physiological measurements (Table 1), chlorophyll content ranged from 24.5 to 45.9, with the highest values located in the city center and in shady locations, and no trees were found to show signs of damage. F_v_/F_m_ values ranged from 0.69 to 0.83, with 10 trees below 0.78 and presuming evidence of harm: 4, 5, 10, 13, 24, 30, 32, 33, 34 and 48. The flavonoid content was the physiological variable that showed the most variability, with values ranging from 0.62 to 1.92. A certain spatial distribution was observed, with the lowest values in the city center. Only three trees showed relatively higher flavonoid values (4, 30 and 33).

The results of the factor analysis, including the loading values and the uniqueness of each element, are summarized in Table 2. Three factors were selected because they absorbed most of the variance explained by the model: F1 = 23.1%; F2 = 18.2%; and F3 = 13.4%. The elements with the highest loadings for each factor (>0.5) are shown in bold in the table. F1 was dominated by Pb and to a lesser extent by Sb, V and Ni; F2 was clearly dominated by Pd and Rh; and F3 by Cu, Hg and again Ni. Uniqueness means the variance that is unique to a specific item and not shared with other items. The predominating items for each factor had lower values of uniqueness (e.g., Pb, Pd and Rh), and the higher the uniqueness, the lower the relevance of the item in the factor model.

Spearman’s correlations between the factor scores of each observation and the physiological variables did not seem to show any pattern (Figure 3A). While the chlorophyll content had a significant positive correlation with F1 (rho = 0.36), flavonoid content was negatively correlated with F1 and F3 (rho = −0.46 and −0.59, respectively), and F_v_/F_m_ was positively correlated with F1 and F3 (rho = 0.35 and 0.39, respectively) but negatively correlated with F2 (rho = −0.28). This last trend was also shown by N (rho= 0.27, 0.69 and −0.24, respectively). Lastly, δ^15^N had positive correlations with F1 and F3 (rho= 0.27 and 0.63). These correlations were summarized in the PCA performed (Figure 3B). The variance collected by the first two dimensions of PCA was 60%. The variables that contributed most to dimension 1 were flavonoid content > N > δ^15^N > F_v_/F_m_ > F3 > F1 > chlorophyll content > F2 > F4. Regarding dimension 2, chlorophyll content > F3 > F1 > F4 > N > F_v_/F_m_ > δ^15^N > flavonoid content. Interestingly, F2 (which was dominated by the PGEs) correlated negatively with the remaining variables, as they pointed to the opposite side of the graphic. The three physiological variables indicating vitality (chlorophyll, F_v_/F_m_ and N content) were significantly and positively correlated between them, but negatively correlated with the physiological variable, usually increasing under adverse factors (flavonoids).

## 4. Discussion

The physiological variables indicating vitality (chlorophyll, F_v_/F_m_ and N content) that were measured in privet trees showed relatively high values, typical of healthy plants [22,23,24,34]. Thus, they did not seem to be affected by heavy metal and PGE pollution in the study area. In fact, they showed significant positive correlations with some pollutants, although the rho values were not too high (Figure 3A). For example, the three vitality variables were positively correlated with F1, which was dominated by heavy metals (mainly Pb, but also Sb, V and Ni). Although the origin of heavy metals in urban environments is mixed in comparison with the more specific sources of metals that can be found in other environments, it seemed that metals associated with F1 had a common source, which could be motor traffic. In this sense, the Pb source could be fuel combustion, although Pb emissions from vehicles have been reduced since the introduction of Pb-free fuels and could also be related to specific industrial emissions; Ni and V are usually attributed to fuel oil and petroleum coke combustion; and Sb is usually attributed to brake and wheel abrasion emissions from roads (data from the website of the Ministry for Ecological Transition and Demographic Challenge of Spain: https://www.miteco.gob.es (accessed on 27 August 2023)). 

Regarding F_v_/F_m_, this variable was positively correlated with F1 and F3, which were the factors dominated by heavy metals from vehicles (as explained above) and negatively correlated with F2, which was dominated by PGEs. This means that PGE pollution could reduce the F_v_/F_m_ values. However, the 20% of trees showing F_v_/F_m_ values below 0.78 (Appendix A), which is the damage threshold below which plant vitality is affected [34], did not coincide with those showing contamination by PGEs. Thus, F_v_/F_m_ only showed a modest response to the pollution levels found in this study, and other fluorescence parameters should be tested in the future for a better interpretation of the effects of pollution on the photosynthetic performance of the plant. The same result as for F_v_/F_m_ was found for N content and the isotopic ratio δ^15^N: positive correlations with heavy metals (F1 and F3) and (only for N content) negative correlations with PGEs (F2). All this suggests that the PGEs, Pd and Rh, behaved differently from the remaining elements, even though they could have had a common source of emission (traffic). This could be due to the fact that Pd and Rh have similar physical and chemical properties, are found together in nature and are used with a specific function as catalytic converters in vehicles [7].

On the other hand, the flavonoid content was negatively correlated with heavy metals (F1 and F3), and the only trees showing high flavonoid values were 4, 30 and 33, which did not show contamination for any of the elements studied (Figure 2). This seems to contradict the role of flavonoids as indicators of abiotic stress, including heavy metal stress [28,29,30]. However, it should be taken into account that, as occurred with the vitality variables, the pollution levels found in the city studied were modest, and thus the plant responses were only slight. Flavonoid content showed a certain spatial distribution, with the lowest values in the center of the city (Figure 1, Appendix A). Probably, this was not due to a real pollution gradient but to other factors, such as the lower radiation received by the trees in the narrower streets of the city center. In fact, flavonoid content can be influenced by many different internal and environmental factors [27,28,29,30,31,32,33,34]. In particular, the flavonoid index measured by the Dualex instrument is especially related to flavonols, which in turn are mostly dependent on UV radiation [31]. Thus, the sun exposure (and UV exposure) of the trees used should be measured in future studies to have a more complete picture of the variability of the flavonoid content. In addition, the sun exposure of the trees could partly be explained by the street width and orientation, the building height, etc., which should also be measured. 

The results obtained could mainly be explained by the fact that there were not many contaminated trees for the elements studied and that the pollutant content found in this work did not seem to be very high (Figure 2). In fact, the metal content (e.g., Cd, Cu, Mn, Pb and Zn) found in privet leaves in the scarce literature available [12,15] was similar or slightly lower than in the present study (see Table 1). Thus, the low pollution levels typical of the study area [35,36] did not influence the physiology of the plant. In contrast, higher levels of pollutants (for example, Cd) can lead to clear responses in the physiological variables measured, with decreases in chlorophyll and F_v_/F_m_ and increases in flavonoids [37]. Similar damage responses were found in some trees in our study, but they could have been due to other types of pollution or environmental stressors. In fact, a study carried out in this same urban area with Mosspheres [36] demonstrated a spatial pattern of the pollution caused by polycyclic aromatic hydrocarbons (PAHs), but this pattern did not coincide with the distribution of heavy metals and PGEs in the trees used in the present study. 

Moreover, factors other than the pollution itself, such as the prevailing wind circulation, should also be taken into account in the interpretation of our results. For example, the prevailing westerly wind in the study area could transport the pollutants generated by the city traffic to the east, contributing to enriching the biomonitor leaves in this zone. This could be the case for Ni (Figure 2), but not for other elements. Thus, the influence of wind in the dispersion of pollutants could be reduced in cities, where buildings of different heights and streets of different widths could greatly modify the wind direction. In this sense, even the way in which pollutants are dispersed should be considered, because (for example) particles would stay closer to the emission source than gaseous pollutants. Another factor influencing the physiological variables measured could be the exposure to solar radiation, because higher exposures generally lead to low chlorophyll content and F_v_/F_m_ values, together with high flavonoid content [22,28]. Finally, trees may also take up pollutants through their roots, being then subsequently transported to the leaves or other parts of the tree. This could mean that the amount of pollutant measured in the leaves might not reflect the real atmospheric pollution. However, Refs. [38,39] measured several heavy metals (Cu, Cr, Mn, Pb and Zn) in different compartments, such as soil and tree leaves, in urban environments, showing that the metals present in leaves came mainly from the atmosphere and not from the soil.

## 5. Conclusions

Our results show that the physiological variables measured in *Ligustrum lucidum* leaves (chlorophyll, F_v_/F_m_, N and flavonoids) responded only slightly to heavy metals and PGEs, probably because of the modest pollution levels in the study area. In addition, these variables could have responded to other environmental factors, masking their specific responses to pollutants. Specifically, microclimatic conditions (sun exposure, water availability, soil properties, etc.) should be considered in future studies for a better interpretation of the results obtained. Even so, the physiological variables used have been useful as biomarkers of the negative physiological effects due to pollution, and thus it is recommended to continue studying these variables in future studies on air quality biomonitoring in urban environments and, in particular, using *L. lucidum*. On the other hand, before starting a study to biomonitor air quality by means of urban vegetation, it should be considered that the sampling network will depend on the presence of the chosen species, so that part of the possible pollution pattern could be missed, consequently influencing the final results. Once this limitation has been considered, city trees should be used to obtain representative air quality maps in urban areas, in order to have information not only on the pollution levels but also on their biological effects. Specifically, *Ligustrum lucidum* can be especially recommended for this use because of its apparent tolerance to low pollution levels, on the basis of the generally high vitality that has been found in our study, with high values of chlorophyll, F_v_/F_m_ and N. Consequently, *Ligustrum lucidum* may provide a good ecosystem service to remove heavy metals and PGEs from the air without compromising its physiological vitality.

## Figures and Tables

**Figure 1 biology-12-01180-f001:**
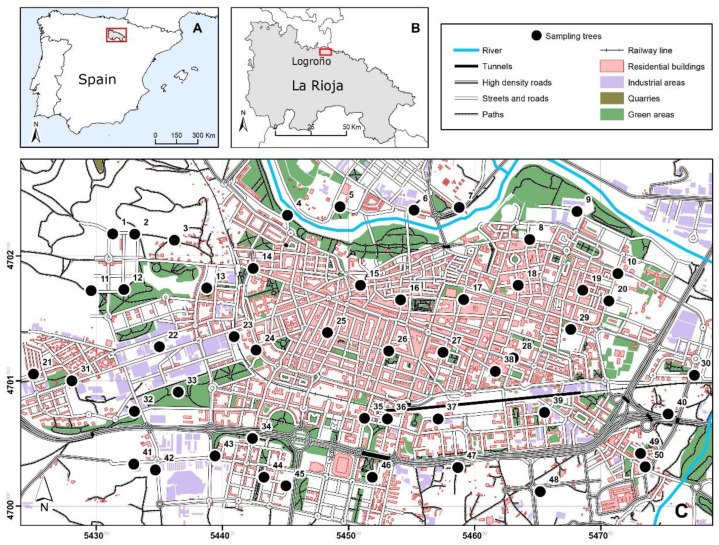
Location of the province (La Rioja) and the city (Logroño) where the study took place in Spain (**A**,**B**). Detail of the map showing the structure of the city and the location of the trees. The sampling grid consisted of 50 trees distributed in a 500 m grid (**C**).

**Figure 2 biology-12-01180-f002:**
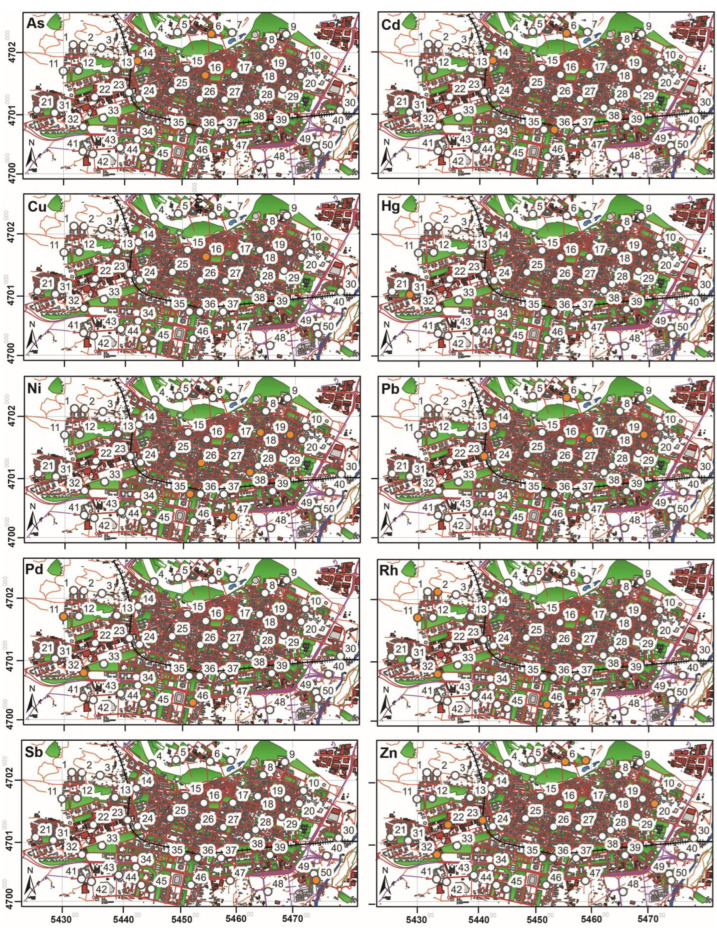
Maps of the city of Logroño for each of the metals and PGEs determined, on the privet leaves of the 50 trees sampled. Unpolluted (white dots) and polluted trees (orange dots) are shown according to the Gaussian mixture model.

**Figure 3 biology-12-01180-f003:**
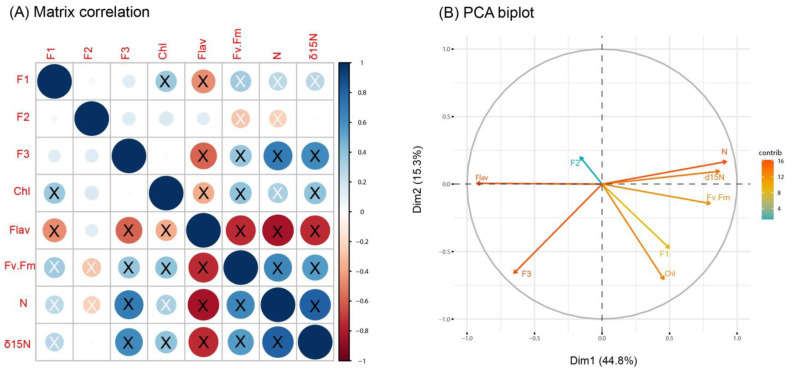
(**A**) Matrix of correlation coefficients between factor scores of each observation and the physiological variables (Chl = chlorophyll, Flav = flavonoids, F_v_/F_m_, N and isotope δ^15^N) analyzed in privet leaves. Positive correlations are displayed in blue and negative correlations in red color. Color intensity and the size of the circle are proportional to the correlation coefficients. X in black = significant correlation *p* < 0.05. X in white = significant correlation *p* < 0.001. (**B**) Principal component analysis (PCA) performed using the 3 factors and the physiological variables. Positively correlated variables point to the same side of the graph, whereas negatively correlated variables point to opposite sides.

**Table 1 biology-12-01180-t001:** Minimum (Min), maximum (Max) and median concentrations, together with the mean absolute deviation (MAD) values, of each pollutant (ng g^−1^ except Cu, Mn and Zn in µg g^−1^). Physiological variables are also shown (index values), as well as N concentrations and nitrogen isotope ratios.

Variable	Min	Max	Median	MAD
As	35.01	181.72	92.44	49.55
Cd	3.26	123.88	23.66	9.93
Cu	3.09	19.81	8.74	2.60
Hg	5.36	34.74	15.66	3.82
Mn	34.68	175.84	71.16	24.99
Ni	88.11	1104.77	356.18	124.63
Pb	87.70	435.65	224.43	59.93
Pd	35.74	662.18	119.05	32.13
Rh	1.09	14.24	2.86	0.67
Sb	18.47	217.03	74.70	28.02
V	71.11	226.87	154.36	26.79
Zn	21.01	153.93	52.77	15.15
Chlorophyll	24.49	45.86	36.45	3.29
Flavonoids	0.62	1.92	1.21	0.25
F_v_/F_m_	0.69	0.83	0.81	0.01
N (%)	0.76	2.78	1.81	0.44
δ^15^N	−1.47	12.21	7.06	2.09

**Table 2 biology-12-01180-t002:** Factor loadings (F1 to F3) calculated with the elements’ levels in the privet leaves. It is considered that a loading value > 0.5 (in bold) indicates that this element is well explained by this factor. Uniqueness of each element is also shown.

Element	F1	F2	F3	Uniqueness
As	0.427	−0.057	−0.093	0.742
Cd	0.349	−0.159	−0.191	0.785
Cu	0.238	−0.395	**−0.560**	0.353
Hg	0.467	0.105	**0.695**	0.274
Mn	0.356	−0.251	−0.096	0.630
Ni	**0.506**	−0.327	**−0.672**	0.000
Pb	**0.905**	0.055	0.129	0.130
Pd	−0.195	**−0.909**	0.294	0.048
Rh	−0.185	**−0.928**	0.320	0.000
Sb	**0.673**	0.086	0.166	0.509
V	**0.673**	−0.051	0.258	0.478
Zn	0.130	−0.329	0.055	0.716

## Data Availability

The raw data presented in this study are available in Appendix A.

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
