# Peer review of "Tree Physiological Variables as a Proxy of Heavy Metal and Platinum Group Elements Pollution in Urban Areas"

_biology, 2023, doi:10.3390/biology12091180_

Round 1
Reviewer 1 Report
This paper analyzed the phsiological parameters of Ligustrum lucidum trees, aiming to make these parameters proxy of the air pollution. I think the authors should reconsider the purpose. Currently the measurement of air pollution is quite easy. I do not see any necessity of using other indirect index to reflect air pollution. Further, the physiological parameters are influenced by many environmental facors, which is also found in the current study. It is bascially impossible to establish a sound correlation under the reallistic ondition. As far as I am concerned, a better way to arrange the research content is to investigate the tolerance or accumulation pattern of this tree to air pollution. Another way is to identify a plant species that is super sensitive to air pollution or a specific pollutant, which may be used to identify pollution.
Any information of the soil pollution, and its correlation with those physiological activities?
Fig2: It may be better to change the presentation way. Can you do the interpolation map?
Author Response
Author's Reply to the Review Report (Reviewer 1)
This paper analyzed the phsiological parameters of Ligustrum lucidum trees, aiming to make these parameters proxy of the air pollution. I think the authors should reconsider the purpose. Currently the measurement of air pollution is quite easy. I do not see any necessity of using other indirect index to reflect air pollution. Further, the physiological parameters are influenced by many environmental facors, which is also found in the current study. It is bascially impossible to establish a sound correlation under the reallistic ondition. As far as I am concerned, a better way to arrange the research content is to investigate the tolerance or accumulation pattern of this tree to air pollution. Another way is to identify a plant species that is super sensitive to air pollution or a specific pollutant, which may be used to identify pollution.
- The use of plants as biomonitors to determine air pollution has been used successfully for decades. However, if we only determine the bioconcentration of pollutants or their accumulation pattern, there is no way to know the possible toxicological effects of pollution on these organisms. And it is impossible to take appropriate measures to assess, manage and mitigate air pollution if we do not know the effects of this pollution. The determination of physiological variables of plants informs us about their vitality, i.e. their state of health and the effects of possible pollution around them, and can therefore be used as a tool for air quality assessment and monitoring to ensure human and ecosystem health. Unfortunately, in the present study, the level of heavy metal pollution is not very high, and the physiological variables seem to be more influenced by other factors; however, this fact does not invalidate their use as a proxy for atmospheric pollution and a tool for assessing and monitoring air quality for human and ecosystem health. Furthermore, this study is one of the first to directly relate the concentration of pollutants in leaves to physiological variables.
All these reasons lead us to think that we do not we should not reconsider our purpose.
Any information of the soil pollution, and its correlation with those physiological activities?
- We agree with the reviewer's suggestion and believe that showing soil data could help to enrich the study, but unfortunately at the time of sampling no soil samples were collected. Nevertheless, and according to the literature as mentioned at the end of the discussion (lines 308-314), it has been shown that in urban environments, metals in leaves come mainly from the atmosphere and not from the soil. Therefore, we suspect that the conclusions of the study would not have changed with soil data.
Fig2: It may be better to change the presentation way. Can you do the interpolation map?
- We assume that by interpolation map the reviewer is referring to why we do not “kriging”. As can be seen in the data there is practically no pollution, so there is no spatial structure for any of the elements, so we believe that statistically it does not make sense to make the interpolation map.

Reviewer 2 Report
Dear Authors
I hope this letter finds you well. I would like to praise you on the thorough research presented in your manuscript titled "Tree physiological variables as a proxy of heavy metal and platinum group elements pollution in urban areas". Your study on the physiological responses of Ligustrum lucidum leaves to heavy metals and PGEs pollution is certainly a significant contribution to the field of environmental science and urban ecology. I appreciate the opportunity to review your work and provide feedback for its improvement.
I have carefully reviewed the manuscript and found it to be well-structured and supported by robust data. However, I believe that a few minor corrections and suggestions for improvement could enhance the clarity and impact of your findings. Please consider the following points:
1. Further Discussion on Flavonoid Content: The study acknowledges the relatively high variability in flavonoid content but provides limited interpretation for this finding. It would be valuable to discuss potential reasons for the observed spatial distribution of flavonoid content and explore the relationship between flavonoid accumulation and specific environmental factors.
2. Consideration of other biotic and abiotic Factors: While the study extensively investigates the impact of pollution on physiological variables, it would be thoughtful to acknowledge and discuss other potential confounding factors that might influence the observed plant responses, such as microclimate variations, water availability, and soil conditions. Experimental work can be considered in future studies.
3. Selection Process for Sample Trees: How was the selection process for the 50 trees used in the study conducted? Were any specific criteria applied to ensure a representative sample?
4. Refinement of Conclusion: The conclusion section could be refined to briefly summarize the key findings and their implications. Emphasize the study's contributions to the field and outline potential avenues for future research based on the obtained results.
Overall, the manuscript represents a valuable contribution to the field of environmental science and urban ecology. With the incorporation of the suggested improvements, the manuscript has the potential to offer a well-rounded exploration of the physiological responses of Ligustrum lucidum leaves to heavy metals and PGEs pollution, enhancing its scientific impact.
General Observation:
There are some grammatical errors and complex sentences that need to be addressed to enhance the clarity and readability of the paper. I recommend carefully reviewing the manuscript to ensure accurate grammar and punctuation, as well as considering breaking down some of the longer sentences to improve understanding.
Author Response
Author's Reply to the Review Report (Reviewer 2)
I hope this letter finds you well. I would like to praise you on the thorough research presented in your manuscript titled "Tree physiological variables as a proxy of heavy metal and platinum group elements pollution in urban areas". Your study on the physiological responses of Ligustrum lucidum leaves to heavy metals and PGEs pollution is certainly a significant contribution to the field of environmental science and urban ecology. I appreciate the opportunity to review your work and provide feedback for its improvement. I have carefully reviewed the manuscript and found it to be well-structured and supported by robust data. However, I believe that a few minor corrections and suggestions for improvement could enhance the clarity and impact of your findings. Please consider the following points:
- Further Discussion on Flavonoid Content: The study acknowledges the relatively high variability in flavonoid content but provides limited interpretation for this finding. It would be valuable to discuss potential reasons for the observed spatial distribution of flavonoid content and explore the relationship between flavonoid accumulation and specific environmental factors.
- Thank you very much for this comment. As the reviewer says, we detected a relatively high variability in flavonoid content in our study. Unfortunately, comparative data from other cities are not available, and this limits the discussion we can offer regarding this variability. In addition, our study was performed in spring and we do not know the overall variability of the flavonoid content over the annual cycle, which again limits the scope of our discussion. On the other hand, flavonoid content can be influenced by many different internal and environmental factors, which have been mentioned in our manuscript (light, UV radiation, nutrients, injuries, herbivores, etc.). This influence has been supported by four selected references (27-30). In particular, the flavonoid index measured by the Dualex instrument is especially related with flavonols, which in turn are mostly dependent on UV radiation (see for example Del Castillo Alonso et al. 2015 in the References). This is to say that a further discussion on flavonoid content should be based on the specific measurement of the sun exposure of each tree (using a radiometer, UV sensors, etc.). Moreover, sun exposure can be influenced by street width and orientation, building height, etc. Unfortunately, we did not measure these variables in our study, although future studies should consider them. Taking into account all these considerations, we have added a short further discussion on flavonoid content at the end of the third paragraph of Discussion.
- Consideration of other biotic and abiotic Factors: While the study extensively investigates the impact of pollution on physiological variables, it would be thoughtful to acknowledge and discuss other potential confounding factors that might influence the observed plant responses, such as microclimate variations, water availability, and soil conditions. Experimental work can be considered in future studies.
- The reviewer is right. The selected physiological variables can be influenced by many different factors (including pollution), and this has been mentioned several times in the manuscript. In addition, a short further discussion has been added on the flavonoid content (see our reply to the previous comment). To make this point more clear, we have specified these considerations in the first sentences of the Conclusions.
- Selection Process for Sample Trees: How was the selection process for the 50 trees used in the study conducted? Were any specific criteria applied to ensure a representative sample?
- The selection process was based on two main points: 1) to design a regular sampling grid; and 2) individuals had to be healthy and robust. This was briefly explained in section 2.1.
- Refinement of Conclusion: The conclusion section could be refined to briefly summarize the key findings and their implications. Emphasize the study's contributions to the field and outline potential avenues for future research based on the obtained results.
- Thanks for this comment. The Conclusion section has been modified to introduce the reviewer’s suggestions (see comment 2).
Overall, the manuscript represents a valuable contribution to the field of environmental science and urban ecology. With the incorporation of the suggested improvements, the manuscript has the potential to offer a well-rounded exploration of the physiological responses of Ligustrum lucidum leaves to heavy metals and PGEs pollution, enhancing its scientific impact.
Comments on the Quality of English Language:
There are some grammatical errors and complex sentences that need to be addressed to enhance the clarity and readability of the paper. I recommend carefully reviewing the manuscript to ensure accurate grammar and punctuation, as well as considering breaking down some of the longer sentences to improve understanding.
- We have tried to improve the English in the revised version of the manuscript.

Reviewer 3 Report
Dear Editor,
Thank you for the opportunity to review this manuscript.
General Comments:
The manuscript is well-written, engaging, and offers significant scientific insights. I do not have any concerns regarding its overall structure or scientific significance.
Specific Concerns:
My primary reservation centers on the authors' choice of fluorescence parameters to characterize stress or tolerance. Specifically, the emphasis on the Fv/Fm parameter raises questions:
The Fv/Fm parameter, while acknowledged in the literature as an indicator of stress, has been reported to only respond after severe stress, suggesting its limited sensitivity to moderate stress.
Potential variables, such as developmental variations or prevailing weather conditions during measurement, can cause fluctuations in Fv/Fm. When relying solely on a parameter like Fv/Fm, artifacts may result, particularly because of the parameter's dependence on Fo (whose influence on chlorophyll content) and potential contributions from PSI fluorescence emission to Fo. These factors could easily lead to misinterpretations.
Considering these implications, I recommend that the authors expand their data set by incorporating other parameters, such as effective quantum yield and photochemical and non-photochemical quenching from PAM measurements, to present a more holistic view.
Recommendations for the Discussion Section:
While I concur that the basal parameters offer invaluable insights into the photosynthetic apparatus's status, the authors would benefit from a more critical evaluation in the discussion, especially concerning chlorophyll fluorescence parameters, with a little dose of self-criticism.
I believe that addressing this central issue will significantly enhance the manuscript's overall quality and clarity.
Author Response
Author's Reply to the Review Report (Reviewer 3)
Dear Editor,
Thank you for the opportunity to review this manuscript.
General Comments:
The manuscript is well-written, engaging, and offers significant scientific insights. I do not have any concerns regarding its overall structure or scientific significance.
Specific Concerns:
My primary reservation centers on the authors' choice of fluorescence parameters to characterize stress or tolerance. Specifically, the emphasis on the Fv/Fm parameter raises questions:
The Fv/Fm parameter, while acknowledged in the literature as an indicator of stress, has been reported to only respond after severe stress, suggesting its limited sensitivity to moderate stress.
Potential variables, such as developmental variations or prevailing weather conditions during measurement, can cause fluctuations in Fv/Fm. When relying solely on a parameter like Fv/Fm, artifacts may result, particularly because of the parameter's dependence on Fo (whose influence on chlorophyll content) and potential contributions from PSI fluorescence emission to Fo. These factors could easily lead to misinterpretations.
Considering these implications, I recommend that the authors expand their data set by incorporating other parameters, such as effective quantum yield and photochemical and non-photochemical quenching from PAM measurements, to present a more holistic view.
- The referee is right because other additional fluorescence parameters could have been used to have a more complete picture of the influence of pollution on the physiology of the plant and, particularly, on its photosynthetic performance. Nevertheless, in our experimental design, we decided to use Fv/Fm because this parameter had already been used as a pollution indicator, and we did not a priori know the pollution levels and the consequent influence on Fv/Fm. In addition, from a practical point of view, we only measured Fv/Fm to be able to take all the physiological measurements as soon as possible, on the same day of collection. Overall, we agree that other fluorescence parameters (which could be more pollution-sensitive than Fv/Fm) should be used in the future for a better interpretation of the results obtained, and this point has been added to the second paragraph of the Discussion.
Recommendations for the Discussion Section:
While I concur that the basal parameters offer invaluable insights into the photosynthetic apparatus's status, the authors would benefit from a more critical evaluation in the discussion, especially concerning chlorophyll fluorescence parameters, with a little dose of self-criticism.
I believe that addressing this central issue will significantly enhance the manuscript's overall quality and clarity.
- See our previous reply.

Round 2
Reviewer 1 Report
The authors have well addressed my concerns. I suggest accept.